MicroRNA biogenesis pathway genes polymorphisms and cancer risk: a systematic review and meta-analysis

He Jieyu 1
Zhao Jun 2 3
Zhu Wenbo 1
Qi Daxun 2
Wang Lina 1
Sun Jinfang 1
Wang Bei 1
Ma Xu 2 3
Dai Qiaoyun 2
Yu Xiaojin 101006181@seu.edu.cn 1
1 Southeast University, Department of Public Health , Nanjing , China
2 National Research Institute for Family Planning , Beijing , China
3 Graduate School of Peking Union Medical College , Beijing , China
Feng Rui
Electronic publication date: 2016 Dec 7
Publication date: 2016
Volume: 4
Electronic Location ID: e2706
Received 2016 Jul 15; Accepted 2016 Oct 20
Copyright: ©2016 He et al.
Copyright year: 2016
Copyright holder: He et al.
License: This is an open access article distributed under the terms of the Creative Commons Attribution License, which permits unrestricted use, distribution, reproduction and adaptation in any medium and for any purpose provided that it is properly attributed. For attribution, the original author(s), title, publication source (PeerJ) and either DOI or URL of the article must be cited.
License URL: https://creativecommons.org/licenses/by/4.0/

Keywords: Meta-analysis, MicroRNA biogenesis, Cancer risk

Funding: National Population and Reproductive Health Information Center National Natural Science Foundation of China 81673274 The study was funded by National Population and Reproductive Health Information Center, National Natural Science Foundation of China (81673274). The funders had no role in study design, data collection and analysis, decision to publish, or preparation of the manuscript.

==============================
MicroRNAs (miRNAs) may promote the development and progression of human cancers. Therefore, components of the miRNA biogenesis pathway may play critical roles in human cancer. Single nucleotide polymorphisms (SNPs) or mutations in genes involved in the miRNA biogenesis pathway may alter levels of gene expression, affecting disease susceptibility. Results of previous studies on genetic variants in the miRNA biogenesis pathway and cancer risk were inconsistent. Therefore, a meta-analysis is needed to assess the associations of these genetic variants with human cancer risk. We searched for relevant articles from PubMed, Web of Science, CNKI, and CBM through Jun 21, 2016. In total, 21 case-control articles met all of the inclusion criteria for the study. Significant associations were observed between cancer risk and the DGCR8polymorphism rs417309 G >A (OR 1.22, 95% CI [1.04–1.42]), as well as the DICER1 polymorphism rs1057035 TT (OR 1.13, 95% CI [1.05–1.22]). These SNPs exhibit high potential as novel diagnostic markers. Future studies with larger sample sizes and more refined analyses are needed to shed more light on these findings.

Introduction

MicroRNAs (miRNAs) are single-stranded RNAs of 20–30 nt that mediate post-transcriptional gene silencing. Over the past decade, it has been reported that miRNAs play key roles in the initiation and progression of human cancer. For instance, the let-7 family of miRNAs can target oncogenes, such as MYC and RAS family members, to inhibit tumor growth (Johnson et al., 2005; Kim et al., 2009). Osada & Takahashi (2011) reported that the miR-17–92 cluster interacts with E2F1 and MYC to promote tumor development in lung cancer. Although individual miRNAs can have either tumorigenic or tumor-suppressive functions, Lu et al. (2005) analyzed 217 mammalian miRNAs from 334 samples and found that miRNAs were globally down-regulated in cancers. Thomson et al. (2006) indicated that this widespread down-regulation of miRNAs is due to a failure in a Drosha processing step, suggesting that miRNA biogenesis may be impaired in cancer.

The majority of the miRNA biogenesis pathway is shown in Fig. 1. In the nucleus, microRNA genes are transcribed as primary miRNAs (pri-miRNAs) by RNA polymerase II (Pol II). These long pri-miRNAs are then cleaved by the double-stranded RNaseIII enzyme Drosha and its essential cofactor DGCR8 (Denli et al., 2004; Lee et al., 2003), generating pre-miRNAs (∼65 nt). Pre-miRNAs are then exported from the nucleus to the cytoplasm by Exportin-5 (XOP5) and its cofactor Ran (Yi et al., 2003). Following nuclear export, the pre-miRNA loop is cleaved by the RNaseIII enzyme Dicer to generate miRNA duplexes (Park et al., 2011). Dicer, TRBP or PACT, and Argonaute proteins Ago1–4 regulate the processing of pre-miRNA and the assembly of the RNA-induced silencing complex (RISC) (Chendrimada et al., 2005). Together with members of the GW182 family of proteins, one strand of the RNA duplex remains anchored to the Ago protein as the mature miRNA while the other one degraded (Gregory et al., 2005).

Figure 1 MiRNA biogenesis pathway.

As mentioned in Fig. 1, the Drosha, DGCR8, Exportin-5, Ran, and Dicer proteins are crucial components in the miRNA biogenesis pathway. Dysregulation of these miRNA-processing machinery components may disturb the product, leading to disease thus this study was focused on this five core components in miRNA biogenesis pathway and explore its link with cancer risk. Indeed, the expression levels of Drosha and Dicer are down-regulated in ovarian cancer (Kobel, Gilks & Huntsman, 2009) and neuroblastomas (Lin et al., 2010), while DGCR8 is up-regulated in esophageal cancer (Sugito et al., 2006), bladder cancer (Catto et al., 2009), and prostate cancer. Exportin-5 is also down-regulated in bladder cancer (Catto et al., 2009). There are several studies implicating single-nucleotide polymorphisms (SNPs) present in those five miRNA biogenesis pathway genes were link with cancer risk, e.g., rs10719 T>C in DROSHA; rs417309 G>A and rs1640299 T>G in DGCR8; rs11077 in XPO5; rs14035 C>T and rs3803012 A>G in RAN; and rs1057035 T>C, rs3742330 A>G, and rs13078 T>A in DICER1. However, the conclusions of previous studies remain inconsistent. A comprehensive review and meta-analysis of SNPs in miRNA biogenesis pathway components is needed. In this study, a meta-analysis was conducted to evaluate the association between SNPs in five genes (DROSHA, DGCR8, XPO5, RAN, and DICER1) involved in the canonical microRNA biogenesis pathway and human cancer risk.

Materials and Methods

Identification of eligible studies

Two reviewers (He and Zhu) searched the electronic databases PubMed, Google Scholar, CBM, and CNKI for studies containing at least one of the five gene names and its aliases (“DROSHA ETOHI2 HSA242976 RANSE3L RN3 RNASE3L RNASEN,” “DGCR8 C22orf12 DGCRK6 Gy1 pahsa,” “XPO5 exp5,” “RAN ARA24 Gsp1 TC4,” or “DICER DCR1 Dicer1e HERNA K12H4 8-LIKE MNG1 RMSE2”). The following terms were also used in each search: cancer, carcinoma, neoplasm, tumor, gene name and its aliases, gene polymorphism, allele, and variation. The search was limited to articles published in English or Chinese through Jun 21, 2016.The reference lists of pertinent articles were checked as well. Articles were included in the study if (1) they reported a correlation between a SNP from DROSHA(rs10719), DGCR8(rs417309 or rs1640299), XPO5(rs11077), RAN(rs14035), or DICER1(rs1057035, rs3742330, or rs13078) and cancer risk; (2) they presented a case-control study; and (3) genotype frequencies were available. Articles were excluded if (1) they were review articles or focused on animals; (2) they lacked data necessary for a systematic review; or (3) the control subjects exhibited a departure from Hardy–Weinberg equilibrium (HWE). In the case where there was more than one publication using the same samples, the article with the most recent publication date was selected. The two reviewer reached consensus on each study.

Data extraction

The following information were extracted by two reviewers (He and Zhao) independently and disagreement were solved by discussion: first author’s name, publication year, country of origin, ethnicity, cancer types, genotype frequencies in case and control groups, HWE results, and source of control samples. Quality assessment of articles was conducted using the Newcastle-Ottawa Quality Assessment Scale (NOS) (Stang, 2010). If more than one type of cancer or multistage research was involved in a single article, data were extracted as independent articles.

Statistical analysis

Hardy–Weinberg equilibrium (HWE) was evaluated in the controls of each SNP using the Chi-square test. Odds ratios (OR) with 95% confidence intervals (CI) were calculated by the Z test to evaluate the association between the SNPs and cancer susceptibility under a dominant model, P < 0.05 was considered statistically significant. The heterogeneity between articles was examined using a Q-test and I2 index (Higgins & Thompson, 2002). Random-effects models (Dersimonian & Nan, 1986) were used if heterogeneity between articles was reported (P < 0.10, I2 > 50%), otherwise fixed-effects models (Mantel & Haenszel, 1959) were applied. Sensitivity analyses were performed with a leave-one-out method by removing each article one at a time and repeating the analysis. Publication bias was evaluated by Egger’s test and Begg’s test, with a P-value > 0.10 considered evidence for no potential publication bias (Begg & Mazumdar, 1994; Egger et al., 1997). All P value were two sided and all analyses were performed by Stata statistical software (Version 12.0; StataCorp, College Station, Texas USA).

Results

Study characteristics

A total of 832 relevant articles were identified based on our search strategy. Article selection procedures for all SNPs from DROSHA, DGCR8, XPO5, RAN, and DICER1 are shown in a flow chart (Fig. 2). As a result, 22 case-control articles including nine SNPs (rs10719, rs417309, rs1640299, rs11077, rs14035, rs3803012, rs1057035, rs3742330, and rs13078) met the inclusion criteria for this study. Characteristics of the eligible studies are summarized in Table 1. Among the applicable articles, 15 articles were about studies of subjects who were of Asian descent, and seven articles were about studies of subjects who were of Caucasian descent. The analyzed articles for each SNP are shown in Table S1.

Figure 2 Flow diagram of the study selection process.

Table 1 Characteristics of the studies eligible for meta-analysis.

Author	Year	Cancer type	Country	Ethnicity	Controls	Case	Control	Method	Polymorphism site	nos	
Horikawa et al.	2008	renal cell carcinoma	American	Caucasian	PB	279	278	SNPlex	rs417309, rs13078, rs3742330, rs10719, rs14035	8	
Yang et al.	2008	bladder cancer	American	Caucasian	HB	746	746	SNPlex	rs417309, rs1640299, rs13078, rs3742330, rs11077	7	
Ye et al.	2008	esophageal cancer	American	Caucasian	HB	346	346	SNPlex	rs417309, rs1640299, rs13078, rs3742330, rs14035, rs11077	7	
Kim et al.	2010	lung cancer	Korea	Asian	HB	100	100		rs417309, rs1640299, rs13078, rs3742330, rs10719, rs14035	7	
Sung et al.	2011	breast cancer	Korea	Asian	HB	559	567	TaqMan	rs1057035, rs11077	7	
Ma et al.	2012	head and neck cancer	China	Asian	HB	397	900	TaqMan	rs1057035, rs3803012	7	
Chen et al.	2013	cervical carcinoma	China	Asian	HB	1486	1549	TaqMan	rs1057035, rs3803012	7	
Jiang et al.	2013	breast cancer	China	Asian	HB	878	900	TaqMan	rs417309, rs1640299, rs1057035, rs13078, rs10719, rs3803012	7	
Liu et al.	2013	hepatocellular carcinoma	China	Asian	HB	1300	2688	TaqMan	rs417309, rs164029, rs1057035, rs3803012	7	
Slaby et al.	2013	colorectal cancer	Czech	Caucasian	HB	197	202	TLDA	rs1057035	7	
Yuan et al.	2013	bladder cancer	China	Asian	HB	685	730	TaqMan	rs1057035, rs13078, rs3742330, rs10719	7	
Roy et al.	2014	oral cancer	India	Asian	HB	451	452	Taqman	rs14035	7	
Cho et al.	2015	Colorectal Cancer	Korean	Asian	HB	408	400	PCR-RFLP	rs3742330, rs10719, rs14035, rs11077	7	
Martin-Guerrero et al.	2015	Lymphocytic Leukemia	Spanish	Caucasian	HB	123	391	Taqman	rs417309, rs1640299, rs1057035,rs13078, rs10719, rs14035	7	
Xie et al. (2015)	2015	gastric cancer	China	Asian	HB	137	142		rs3742330, rs14035, rs11077	7	
Zhao et al. (2015)	2015	colorectal cancer	China	Asian	HB	163	142		rs3742330, rs14035, rs11077	7	
Zu et al. (2013)	2013	lung cancer	China	Asian	HB	600	600	TaqMan	rs1057035	7	
Zheng Liang (2013)	2013	esophageal cancer	China	Asian	HB	380	380		rs3742330	7	
Zhang (2012)	2012	gastric cancer	China	Asian	HB	1674	1852	TaqMan	rs3803012, rs1057035	7	
Buas et al. (2015)	2015	esophageal cancer	Europe	Caucasian	HB	600	600	TaqMan	rs14035, rs11077	7	
Gutierrez-Camino et al. (2014)	2014	Lymphocytic Leukemia	Spanish	Caucasian	HB	213	387		rs1640299	7	

Quantitative synthesis

Evaluations of the associations between miRNA biogenesis pathway component SNPs and human cancer risk are shown in Table 2 and Fig. 3.

Table 2 Analysis of associations between SNPs from DROSHA, DGCR8, XPO5, RAN, and DICER1 and cancer risk.

Gene(locus)	Position	studies	Method	Cases /controls	WW vs.WM+MMa	Pb	I2	WW vs. WM+MMa	Pc	I2	WW vs.WM+MMa	Pd	I2	
					ORb(95% CI)			ORc(95% CI)			ORd(95% CI)			
DROSHA(5p13.3)														
rs10719 T>C	3′UTR	5	R	1982/2293	0.91(0.75,1.10)	0.209	36.2	1.34(0.78,2.30)	0.070	69.6	1.05(0.83,1.33)	0.026	63.9	
DGCR8(22q11.2)														
rs417309 G>Ae	3′UTR	7	F	3327/3658	1.44(1.13,1.83)	0.561	0.0	1.04(0.78,1.39)	0.147	47.8	1.22(1.04,1.42)	0.190	31.2	
rs1640299 T>G	3′UTR	7	R	2610/3046	1.07(0.90,1.28)	0.571	0.0	1.22(0.85,1.75)	0.017	70.7	1.20(0.94,1.54)	0.038	46.1	
XPO5(6p21.1)														
rs11077 A>G	3′UTR	7	F	8065/5478	0.83(0.66,1.03)	0.652	0.0	0.95(0.88,1.03)	0.260	25.7	0.94(0.87,1.01)	0.463	0.0	
RAN(12q24.3)														
rs14035 C>T	3′UTR	9	R	7702/5335	1.40(0.72,2.69)	0.001	92.6	0.99(0.73,1.33)	0.002	79.2	1.17(0.87,1.57)	0.000	88.7	
rs3803012 A>G	3′UTR	5	F	5642/6489	0.98(0.87,1.11)	0.494	0.0	–	–	–	–	–	–	
DICER1(14q32.13)														
rs1057035 T>C	3′UTR	10	F	7783/8925	1.13(1.04,1.22)	0.163	33.2	1.21(0.90,1.62)	0.182	44.0	1.13(1.05,1.22)	0.188	27.8	
rs3742330 A>G	3′UTR	9	R	3222/3240	1.02(0.89,1.18)	0.000	79.6	0.90(0.73,1.10)	0.396	0.0	0.98(0.87,1.10)	0.001	70.9	
rs13078 T>A	3′UTR	7	F	3102/3419	1.01(0.81,1.26)	0.889	0.0	1.13(0.97,1.32)	0.116	49.3	1.09(0.96,1.24)	0.339	11.8	
Notes.

a W: major allele M: minor allele.

b Asian population.

c Caucasian population.

d All over.

e WM+MM vs. WW.

Method: F, Fixed model; R, Random model.

A significant association was observed between cancer risk and the DGCR8 rs417309 G>A polymorphism in the overall pooled analysis (GA + AA vs. GG: OR 1.22, 95% CI [1.04–1.42]) and in the Asian samples (GA + AA vs. GG: OR 1.44, 95% CI [1.13–1.83]). For the rs1057035 T>C variant in DICER1, a significant (13%) increase in cancer risk was found in the overall pooled analysis (AA vs. AC + CC: OR 1.13, 95% CI [1.05–1.22]). Subgroup analysis determined by sample ethnicity revealed that genotypes containing the major allele increased the risk of cancer in those of Asian descent (AA vs. AC + CC: OR 1.13, 95% CI [1.04–1.22]).

Figure 3 Forest plot for the relationship between the microRNA biogenesis pathway genes polymorphism and cancer risk.

Sensitivity analysis

Sensitivity analysis was conducted to assess the robustness of the results. Each individual article was omitted to measure its effect on the pooled ORs (Fig. S3). The sensitivity analysis forest plot showed that no individual article dramatically affected the pooled OR for any SNP.

Figure 4 Funnel plot for publication bias test. OR, odds ratio; SE, standard error.

Publication bias

Funnel plots did not show obvious asymmetry for any locus (Fig. 4). The results of Egger’s test and Begg’s test revealed that rs10719 in DROSHA presented publication bias (Begg’s test: Z = 1.47, P = 0.14; Egger’s test: P = 0.09, 95% CI [−0.87–7.07]), suggesting that the results for rs10719 should be treated with caution. For the other SNPs, neither test indicated potential publication bias (Table 3).

Discussion

In this study, we searched for articles indicating an association between SNPs in miRNA biogenesis pathway genes and human cancer risk. A total of 21 articles were identified, and 9 SNPs were evaluated for association with cancer susceptibility. The results demonstrated that the GG genotype of rs417309 in DGCR8 was significantly rarer among cases compared with controls in the overall pooled analysis and the TT genotype of rs1057035 in DICER1 was associated with a 13% increase in cancer risk. In addition significant associations were found in the Asian population but not in the Caucasian population, which suggested a possible ethnic difference in the genetic background and the environment. Significant association were not observed in DROSHA (rs10719 T>C), XPO5 (rs11077 A>G) and RAN (rs14035 C>T, rs3803012A>G). Genome-wide association studies (GWAS) involving associations between esophageal adenocarcinoma risk and RAN(rs14035 C>T) or XOP5(rs11077 A>G) were included in this meta-analysis (Buas et al., 2015). The results of our meta-analysis were consistent with the findings of these GWAS publications (OR 1.10, 95% CI [0.76–1.59] compared with OR 1.17, 95% CI [0.87–1.57] in our meta-analysis for RAN rs14035 C>T and OR 0.94, 95% CI [0.85–1.04] compared with OR 0.93, 95% CI [0.87–1.01] in our meta-analysis for XOP5 rs11077 A>G).

Table 3 Results of Egger’s and Begg’s tests for publication bias.

Category	Studies	Begg’s test	Egger’s test	
		Z	P-value	(95%) CI	P-value	
DROSHA						
rs10719 T>C	5	1.47	0.14	(−0.87,7.07)	0.09	
DGCR8						
rs417309 G>A	7	1.20	0.23	(−4.79,0.79)	0.13	
rs1640299 T>G	7	1.20	0.23	(−2.39,5.13)	0.39	
XPO5						
rs11077 A>G	7	0.3	0.76	(−0.20,0.12)	0.53	
RAN						
rs14035 C>T	9	1.15	0.25	(−2.66,5.07)	0.49	
rs3803012 A>G	5	−1.32	0.19	(−3.75,1.80)	0.39	
DICER1						
rs1057035 T>C	10	0.72	0.47	(−2.11,4.20)	0.76	
rs3742330 A>G	9	0.10	0.92	(−2.98,8.01)	0.32	
rs13078 T>A	7	−1.05	0.293	(−3.81,2,91)	0.77	

DROSHA and its essential cofactor DGCR8 are critical protein that executes the initial step in microRNA processing (Lee et al., 2003). Dysregulation of DROSHA and DGCR8 has been observed in many cancers such as epithelial skin cancer (Sand et al., 2010), breast cancer (Chen et al., 2013) and ovarian cancer (Guo et al., 2015). Pre-miRNAs are exported into the cytoplasm to generate mature miRNAs through XOP5 and it cofactor RAN (Lin & Gregory, 2015), some heterozygous XPO5 variations were found in colon, gastric and endometrial cancer (Melo et al., 2010). The variation of XPO5 impairs pre-miRNA exportation and lead to defect in miRNAs biogenesis. Genetic and epigenetic association studies reported that the genetic variation of XPO5 was associated with the risk of breast cancer (Leaderer et al., 2011). DICER1 is another enzymes that plays critical role in the cleavage of pre-miRNAs into their mature form (Lin & Gregory, 2015). The variation of DICER1 were correlated with cancer risk through affecting cell proliferation and apoptosis (Dedes et al., 2011) all those component are key enzymes in miRNAs mature process. Since the global impairment of mature miRNAs is emerging as a common feature of human tumors (Melo et al., 2010), and given the critical functions of Drosha, DGCR8, Exportin-5, Ran, and Dicer in miRNA biogenesis, it is logical to presume that genetic polymorphisms in these genes may affect the processing of miRNAs and, thus, cancer risk. As a matter of fact, several studies have shown significantly association about single nucleotide polymorphism of MicroRNA biogenesis pathway gene with different cancer risk. Cho et al. (2015) reported that RAN rs14035 CT heterozygotes and XPO5 rs11077 AA carriers experienced reduced risk of colorectal cancer in Korean population. Leaderer et al. (2011) performed both genetic and epigenetic association studies of XPO5 in a case-control study of breast cancer and found XPO5 rs11544382 was associated with cancer risk (the analysis of XPO5 rs11544382 with cancer risk was not performed due to lack of availed data). Chen et al. (2013) analyzed 1,486 cervical cases and 1,549 cancer-free controls in Chinese population and found that a single nucleotide polymorphisms in DICER rs1057035 and RAN rs3803012 was associated with cervical cancer risk. A case-control and further research conducted by Jiang et al. (2013) suggested that DGCR8 rs417309 G>A might affect breast cancer risk through interrupting the miRNA binding. However, conclusions of miRNA biogenesis pathway genes polymorphisms and cancer risk remain inconsistent, which may cause by the heterogeneity of the cancer subtype, sample size and the ethnicity of patients. Yu, Kuang & Yin (2015) conducted a meta-analysis synthesizes sevens studies with multi-type cancer and found the C allele of the DICER rs1057035 polymorphism probably decreases cancer risk (We included more studies in our studies and found the same results in DICER rs1057035). This study based on the association of a single SNP or gene with cancer risk, which may weaken gene’s biological value. All five miRNA biogenesis pathway were evaluated in our analysis, which may provide a more comprehensive view of the true system. To the best of our knowledge, this study represents the first meta-analysis to focus on the association between human cancer risk and SNPs across the entire miRNAs biogenesis pathway.

Sensitivity analysis indicated that the results of our study were robust. Egger’s test and Begg’s test revealed that publication bias has little influence on the results, with the exception of rs10719 in DROSHA, suggesting that the results for rs10719 should be considered with caution. Despite these results, some limitations still apply to this meta-analysis. First, all results were based on unadjusted estimates because few adjusted ORs were reported in the included articles. Biases caused by confounding variables such as sex, age, and smoking may therefore persist. Secondly, heterogeneity among different cancers may cause real effects to be hidden when pooling all cancer types. A stratified analysis based on cancer type was not performed due to the lack of an adequate number of publications for a single cancer type. Third, although the results of the publication bias analysis indicated no bias for any locus except rs10719, it is possible that there actually was publication bias but that it was not identified by the Egger’s and Begg’s tests due to the low power given to the small number of studies (Begg & Mazumdar, 1994; Egger et al., 1997). Finally, we applied a traditional fixed model and a random model to pooled multiple-ethnicity samples; neither of these approaches is ideal for capturing the heterogeneity of effects observed across different ethnic populations (Li & Keating, 2014).

In summary, this meta-analysis suggests a potential role of the miRNA biogenesis genes DGCR8 (rs417309 G>A) and DICER1 (rs1057035 T>C) in cancer risk. The identified markers can thus potentially be used as biomarkers for cancer diagnosis. However, future well-designed studies, including those focused on a single cancer type with larger sample sizes, are still needed to confirm these observed associations due to the limitations mentioned above.

Supplemental Information

Supplemental Information 1 PRISMA checklist

Click here for additional data file.

Figure S1 Forest plot of sensitivity analysis

Click here for additional data file.

Table S1 Raw data of each SNP in the meta-analysis

Characteristics of eligible studies for each SNP in the meta-analysis

Click here for additional data file.

Additional Information and Declarations

Competing Interests

Author Contributions

Data Availability

The authors declare there are no competing interests.

Jieyu He performed the experiments, wrote the paper.

Jun Zhao performed the experiments, prepared figures and/or tables.

Wenbo Zhu analyzed the data, prepared figures and/or tables.

Daxun Qi analyzed the data.

Lina Wang reviewed drafts of the paper.

Jinfang Sun analyzed the data, contributed reagents/materials/analysis tools.

Bei Wang contributed reagents/materials/analysis tools, reviewed drafts of the paper.

Xu Ma and Qiaoyun Dai contributed reagents/materials/analysis tools.

Xiaojin Yu conceived and designed the experiments, reviewed drafts of the paper.

The following information was supplied regarding data availability:

The raw data has been supplied as a Supplemental File.

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
