# Peer review of "MicroRNA biogenesis pathway genes polymorphisms and cancer risk: a systematic review and meta-analysis"

_PeerJ, doi:10.7717/peerj.2706_

## Round 0.1 · original submission · Major Revisions

All the major concerns from two reviewers need to be addressed. Let me know if you need more time.

Reviewer 1 ·

Basic reporting

None

Experimental design

None

Validity of the findings

None

Additional comments

I find the paper very interesting. Not many research groups pool published data together to do a meta-analysis though it is crucial to do so in the big data era now. The paper overall is well composed. However, major concerns I have are: (1) the discussion of the results are too brief; (2) no explicit statistical models for other people to reproduce the results. Specifically, when authors discuss table 2, there is no explanation of the column "Pa", "Pb", and "Pc". Are they the p-values associated with the odds ratio test? Then what are the cutoffs do you use to decide significance? Similarly, for each column of the Table 2, the authors should explain what they are, and what they mean, etc. However, in the paper, this is very brief. Secondly, the statistical models are not explained clearly enough that other data scientists can reproduce the work. The authors simply state "fixed effect models" and "random effect models". It would be better if they could write down the actual regression formula to let people know what models they use. Such as, what are the response variables, what are the regressors, any transformations, etc. Since (1) discussion is too brief and (2) no explicit statistical models in the paper, it is hard for me to decide the correctness of the paper. However, I like the overall research idea of mining pooled data from existing publications.

Reviewer 2 ·

Basic reporting

This manuscript describes a meta-analysis for association between general cancer risk and the microRNA biogenesis pathway genes. Literature search was conducted based on 5 microRNA biogenesis pathway related genes, and only cancer case-control studies were included. A small number of articles (n=22) met the inclusion criteria in total, then z-test was performed using odds ratio and CI to evaluate the association between the microRNA biogenesis pathway related SNPs and cancer susceptibility. Funnel test, Egger’s and Begg’s tests were performed for publication bias. In recent years, microRNAs in cancer research is becoming a hot are in biomedical science, which guarantees the impact of this topic. Overall the analyses were conducted well; the manuscript was written and organized very clearly.

Experimental design

However, the study design does not appear to be valid, most likely because of the stratification analysis was not performed to successfully evaluate the roles of these SNPs in each cancer type. Instead, a cancer type pooled analysis was performed considering the insufficient number of literatures. There are 12 cancer types studied in total in only 22 articles (Table 1). Another major concern is the statistical methods or models used in this study were not clearly described. Validation study using public available data from online resource such as TCGA is desirable to support the current conclusion.

Validity of the findings

Included in above area

---

## Round 0.2 · accepted · Accept

All concerns from previous reviewers have been sufficiently addressed. It will be a great addition to PeerJ!

Reviewer 1 ·

Basic reporting

None.

Experimental design

None.

Validity of the findings

None.

Reviewer 2 ·

Basic reporting

No Comments

Experimental design

No Comments

Validity of the findings

No Comments

Additional comments

No Comments